# College openings in the United States increase mobility and COVID-19 incidence

**Martin S. Andersen**[1�ID]*, **Ana I. Bento**[2☺¤]*, **Anirban Basu**[3], **Christopher R. Marsicano**[ID][4,5], **Kosali I. Simon**[6]

**1** Department of Economics, University of North Carolina at Greensboro, Greensboro, North Carolina, United States of America, **2** Department of Epidemiology and Biostatistics, School of Public Health, Indiana University-Bloomington, Bloomington, Indiana, United States of America, **3** The Comparative Health Outcomes, Policy, and Economics (CHOICE) Institute, Departments of Pharmacy, Health Services, and Economics, University of Washington, Seattle, Washington, United States of America, **4** The College Crisis Initiative at Davidson College, Davidson College, Davidson, North Carolina, United States of America, **5** Educational Studies Department, Davidson College, Davidson, North Carolina, United States of America, **6** O'Neill School of Public and Environmental Affairs, Indiana University, Bloomington, Indiana, United States of America

☺ These authors contributed equally to this work.
¤ Current address: Pandemic Prevention Institute, Rockefeller Foundation, New York, New York, United States of America
* msander4@uncg.edu (MSA); abento@iu.edu (AIB)

**Data Availability Statement:** ***PA @ ACCEPT: Please request public OSF link at accept*** All data used can be requested from SafeGraph, the CDC, and the College Crisis Initiative or are publicly available at USAFacts.org. Most of the data

## Abstract

School and college reopening-closure policies are considered one of the most promising non-pharmaceutical interventions for mitigating infectious diseases. Nonetheless, the effectiveness of these policies is still debated, largely due to the lack of empirical evidence on behavior during implementation. We examined U.S. college reopenings' association with changes in human mobility within campuses and in COVID-19 incidence in the counties of the campuses over a twenty-week period around college reopenings in the Fall of 2020. We used an integrative framework, with a difference-in-differences design comparing areas with a college campus, before and after reopening, to areas without a campus and a Bayesian approach to estimate the daily reproductive number ($R_t$). We found that college reopenings were associated with increased campus mobility, and increased COVID-19 incidence by 4.9 cases per 100,000 (95% confidence interval [CI]: 2.9–6.9), or a 37% increase relative to the pre-period mean. This reflected our estimate of increased transmission locally after reopening. A greater increase in county COVID-19 incidence resulted from campuses that drew students from counties with high COVID-19 incidence in the weeks before reopening ($\chi^2(2)$ = 8.9, $p$ = 0.012) and those with a greater share of college students, relative to population ($\chi^2(2)$ = 98.83, $p$ < 0.001). Even by Fall of 2022, large shares of populations remained unvaccinated, increasing the relevance of understanding non-pharmaceutical decisions over an extended period of a pandemic. Our study sheds light on movement and social mixing patterns during the closure-reopening of colleges during a public health threat, and offers strategic instruments for benefit-cost analyses of school reopening/closure policies.

underlying our study cannot be publicly released. The College Crisis Initiative data will be posted online by July 1, 2022 (this is the earliest practical date). Our CDC data cannot be shared publicly because our agreement with the CDC prohibits us from sharing those data. Requests for access to those data can be made via the webform at https://data.cdc.gov/Case-Surveillance/COVID-19-Case-Surveillance-Restricted-AccessDetai/mbd7-r32t. We are also prohibited from sharing our human mobility data, which are available from SafeGraph Inc (www.safegraph.com). COVID-19 incidence data are from USAFacts.org and can be obtained at https://static.usafacts.org/public/data/covid19/covid_confirmed_usafacts.csv. All other data and code are available at https://osf.io/e6ydq/?view_only=63cfb8cb6a2b4f54a5fc66adff659d89. We will provide an unblinded and public link at your request.

**Funding:** The author(s) received no specific funding for this work.

**Competing interests:** The authors have declared that no competing interests exist.

## Introduction

One of the key lessons learned from the COVID-19 pandemic has been the pivotal role of human behavior, specifically mobility and mixing in spreading infection, and the role of young adults. In the United States and globally, these phenomena are acutely important in congregate and communal living settings that are common not only in colleges and prisons but also in nursing homes [1–4]. However, the role of communal living, and its interaction with mobility and mixing, is difficult to identify empirically since people enter communal living settings non-randomly. The resumption of teaching on a college campus provides a sudden change in a community's exposure to communal living and differences across college campuses lead to variation in the extent to which campus reopenings induce mixing with higher and lower incidence areas.

The susceptibility of children and college-age individuals to COVID-19 and their role in transmission has been heavily debated and remains hard to quantify [5–9]. Following the first wave of school closures in the United States in the spring of 2020, COVID-19 incidence fell across the country, leading many public health officials to view closing schools as a viable strategy to mitigate the spread of the pandemic [10, 11]. However, closing schools, while potentially reducing transmission, may adversely affect children and college students. As a result, it is important to understand what role college reopenings play, if any, in the COVID-19 pandemic to design efficient mitigation strategies now and in the future.

During late Summer 2020, colleges and universities across the United States reopened and welcomed hundreds of thousands of students back to campus in the United States [12]. Over half of these institutions reopened for in-person teaching, although many institutions switched to online instruction after rapid increases in reported COVID-19 cases on campuses and in the community [13, 14]. A few studies have sought to formally test the hypothesis that reopening college campuses increased COVID-19 incidence [4, 15–18]. However, the institutions in these studies represent a small proportion of the 11 million undergraduates enrolled in public and non-profit four-year institutions across the country [12]. A phylogenetic study from western Wisconsin [3] identified two clusters of SARS-CoV-2 strains on college campuses that may have subsequently infected nursing home residents, demonstrating transmission between college campuses and the surrounding community. Nonetheless, the effectiveness of college reopening policies as non-pharmaceutical interventions for mitigating the burden of COVID-19 is still disputed. Simulation-based studies have been unable to provide public health officials with conclusive recommendations, despite detailed COVID-19 transmission datasets [19, 20]. The lack of a clear direction is mostly due to insufficient data about the college-specific details and how to harness movement as proxies for behavior and mixing patterns of the population while such strategies are in place. As we approach Fall 2022, with expected mass movement events in the US, millions of college and university students will return to residential instruction. This leaves little time to achieve high levels of full vaccination necessary to prevent outbreaks. Furthermore, due to new variants now circulating, there is an increased risk of breakthrough infections [21]. Thus, it is more important than ever to understand school reopenings' effects and mass mobility events on COVID-19 incidence.

We harnessed comprehensive, national data covering the start date and instructional method of most four-year U.S. colleges and universities together with a highly resolved dataset (both spatially and by age) from the CDC, [22] which provided detailed demographic information on COVID-19 cases around the country. This gave us the ability to directly measure the variation in human movement patterns caused by the policy and, in addition, allowed us to identify college-age cases and assign cases based on symptom onset. We hypothesized that reopening colleges would increase COVID-19 transmission within the college community

with potential spillover effects onto the neighboring populations. We also hypothesized that increases in incidence would be greater on campuses that attract students from areas with a higher incidence of COVID-19 and that these effects would be concentrated among campuses providing face-to-face instruction. While there is some compelling research around testing [23, 24] and limiting student mobility [9] as COVID-19 mitigation strategies, it is outside of the scope of our study to understand the impact of specific actions colleges may have taken in response to rising rates.

We use an integrative framework, with a difference-in-differences design comparing areas with a college campus, before and after reopening, to areas without a campus and a Bayesian approach to estimate the daily reproductive number ($R_t$). We unequivocally demonstrate that there was a marked increase in COVID-19 incidence among college-age students following the reopening of campuses. Finally, while COVID-19 case counts have been a focus of several studies, our data also allowed us to examine other public health outcomes, such as hospitalizations or deaths. Our results provide evidence of the COVID-19 impact of colleges-reopening policies locally and in neighboring areas and undoubtedly informs future events in these settings.

The broad availability of vaccines that prevent the development of COVID-19, but which do not prevent transmission of the underlying virus, highlights the importance of having tools available that can inhibit transmission of the virus. The fact that there are some pockets of the United States and the world with very low vaccination rates also demonstrates the importance of preventing transmission since an outbreak of the virus in such an area will have significantly worse effects on the population than in areas with greater vaccine uptake. College reopening and teaching strategies provide one layer in a series of layers that seek to prevent sustained transmission of SARS-CoV2, the viral cause of COVID-19.

## Materials and methods

### Study data

**College characteristics.** We collected data on opening dates and announced instructional methods from the College Crisis Initiative at Davidson College (C2i) [25] for 1,431 public and non-profit colleges and universities ("colleges") in the United States. The College Crisis Initiative collects data on nearly all non-profit and public four-year degree-granting institutions with full-time undergraduates that receive Title IV aid. It excludes four-year for-profit institutions, specialty institutions like seminaries or stand-alone law schools, or institutions with graduate-only programs. This list comes from the Integrated Postsecondary Education Data System (IPEDS), which lists in total 6,527 institutions ranging from research universities to non-degree-granting institutions like local cosmetology schools. IPEDS indicates that of those, 2,009 are four-year public and non-profit degree-granting institutions with first-time, full-time undergraduates. Our sample, therefore, represents nearly 70 percent of these institutions. Further, this represents 70 percent of total undergraduate enrollment among all institutions of higher education in the United States (author calculations based on IPEDS administrative 2018 data).

We assigned college campuses to Census Block Groups (CBGs) using a college campus shapefile (geographic coordinates) prepared by the Department of Homeland Security [26]. We used a spatial join to assign each Census Block Group to the college campus that occupied the largest area in the block group, as a result, our assignment of campuses to block groups was unique. We then merged these data with college opening dates. Our final sample included 1371 schools in 786 counties. We assigned reopening strategies based on the mode of instruction reported on the date instruction began for Fall 2020. Campuses were classified as in-

person or online based on the instructional modality in effect the day classes resumed for the Fall semester. Institutions that instituted primarily hybrid (379) or primarily in-person (493) modes of instruction were classified as "in person." Institutions that offered only online classes, or for which the majority of the classes offered were online were classified as "online" (499). We assigned instructional modalities to the 786 counties with a college in our sample based on the status of the first campus to reopen in each county and, if necessary, the largest campus of those that opened on the same day. We classified 552 counties as in-person and 234 as online (S1 Table). In the average "In-person" county, 6% of campuses reopened for online teaching, while in the average "Online" county 14% of campuses reopened for in-person teaching. Counties with colleges that were not in our sample were included in the control group.

**Mobility.** We extracted cellular data from SafeGraph's Social Distancing Metrics files. SafeGraph aggregates anonymized location data from numerous applications in order to provide insights about physical places, via the Placekey Community. To enhance privacy, Safe-Graph excludes CBG information if fewer than five devices visited an establishment in a month from a given CBG. These data measure the number of devices that are detected each day in each CBG, from June 24th through November 9th of 2020. SafeGraph data have been used in several recent publications [27–32].

**COVID-19 cases and sequelae.** We used aggregate cumulative case data at the county level from USAFacts and deidentified, case-level data from the Centers for Disease Control to estimate the incidence of COVID-19 in a county by age-group [22]. The University of North Carolina at Greensboro Institutional Review Board reviewed and approved our use of the CDC data and waived the requirement for informed consent on the part of individuals in the dataset. The CDC does not make any claims regarding the accuracy or validity of its data therefore we restricted our use of the CDC data to those counties in which the cumulative number of cases at the end of our study period was no less than 95% of the USAFacts estimate for that same county and the correlation in the rolling thirty day incidence of cases exceeded 0.95. S1 Fig provides a map of the 1917 counties that met our inclusion criteria for the CDC data. Using the CDC data we estimated the number of cases diagnosed in each county, age-group, date cell and the number of cases that were, by March 31, 2021, hospitalized, admitted to the ICU, or resulted in death. We converted these values into values per 100,000 people in a age-county cell using population data from the 5-year American Community Survey [33]. The CDC data identifies the ultimate outcome of cases by diagnosis date, so our data indicates the number of incident cases per 100,000 people and the number that resulted in death, hospitalization, or ICU admission.

In the Additional Methods (S1 Appendix) of the Supplementary Information, we describe our estimation of the effective reproductive number ($R(t)$) and how we constructed our index for college exposure to other counties.

## Statistical analysis

Our main analyses use a panel of counties and Census Block Groups (CBGs). In our county level analyses we identified the earliest and, if necessary, largest college or university in each county and assigned the county that college or university's reopening modality. We estimated generalized difference-in-differences (DiD) models [34] for mobility to campus, COVID-19 incidence, and COVID-19 cases resulting in a hospitalization, ICU admission, or death by March 31 2021. To incorporate variation across age groups, we also used age-specific data on COVID-19 incidence and cases resulting in hospitalization, ICU admission, and death. Analyses of mobility used Census Block Group (CBG) data, while for all other outcomes we used county-level data.

For each reopening date (a group) and calendar date, the generalized DiD model computes a separate estimate of the average effect of treatment on the treated (ATT) using other counties or block groups that did not have a college as controls. In these calculations, the generalized DiD embeds a propensity score estimation and an outcome regression. We model both using the natural log of the population in the county (census block group) and the number of devices visiting K-12 schools, as a proxy measure for school reopening policies that accounts for differences in teaching modalities. We aggregate these ATT estimates into daily and weekly estimates of the average effect of treatment on the treated. Using the weekly aggregate we also tested for pre-trends using $\chi^2$ tests for the joint-significance of the weekly ATTs from 7 to 3 weeks before reopening. We computed an overall DiD estimate for the effect of reopening as the average of the ATTs for the period from day 0 to day 27 following reopening and computed standard errors using a clustered multiplier bootstrap [34].

We also computed estimates for each reopening type comparing each reopening type to counties and CBGs that did not have a college. We then computed cross-type tests for the equality of the DiD coefficients using $\chi^2$ tests after bootstrapping the covariance matrix between outcomes.

We tested for an effect of exposure to students from high-incidence counties by breaking the exposure index into terciles and computing tercile specific DiD estimates and conducted a similar exercise using terciles of the fraction of college students in a county. We then tested for equality across terciles using $\chi^2$ tests.

We also assessed the robustness of our results to allowing for violations of the parallel trends assumption [35] using the HonestDiD package in R and to the existence of multiple colleges in a single county using a subsample of counties that either contained no colleges or contained exactly one college in our sample.

Means and standard deviations of our dependent variables are presented in S2 Table.

## Results

Our study period ran from July 5th 2020 to November 1st 2020, which spanned the four weeks before the first campus reopening and four weeks after the last campus reopened. Of the 3,142 counties of the United States, 784 contained a college campus from our universe of 1,360 colleges. However, over 238.0 million people live in counties with a college campus. S2 Fig maps the campuses in our sample by teaching modality.

Our identification strategy made comparisons between counties with and without a college campus around the time that a campus reopened in a "difference-in-differences" design [36]. Since several counties contained more than one college campus, we assigned county status based on the status of the first, and, as a tie-breaker, largest, campus to reopen in each county.

### Event studies

The reopening of a college affected mixing patterns not only of the students but also the members of the surrounding communities where these students live. The number of devices on campus increased significantly in the week before campuses reopened and remained high for at least the first 28 days following reopening ($\chi^2(28) = 1355.8$, $p < 0.001$, Fig 1a). Aggregating by week, which smooths out day of the week fluctuations in movement, and separating the sample by teaching modality demonstrated that there were significant increases in movement to census block groups containing college campuses after those campuses reopened regardless of the teaching modality (In-person: $\chi^2(8) = 791.03$, $p < 0.001$; Online: $\chi^2(8) = 249.37$, $p < 0.001$ Fig 1b), although the increase was larger for in-person reopenings ($\chi^2(8) = 330.8$, $p < 0.001$). The increase in mobility was accompanied by a rise in COVID-19 incidence,

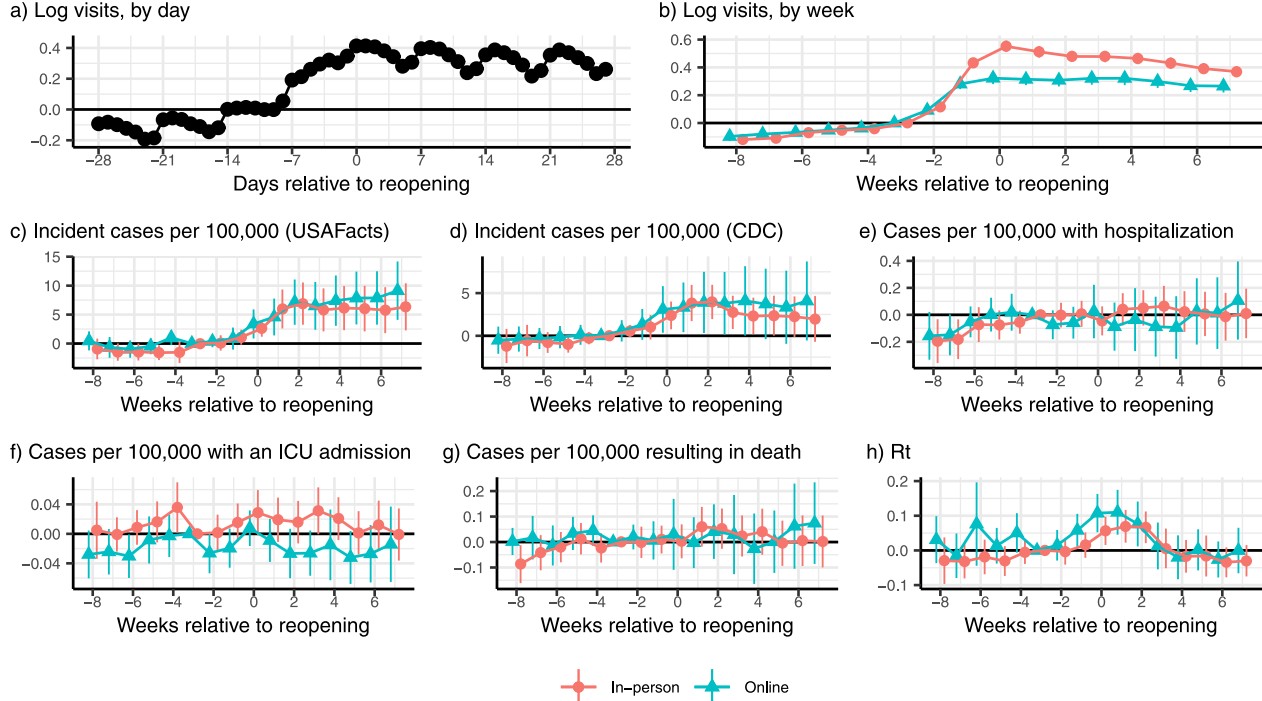

**Fig 1. Event study estimates of reopening college campuses, relative to counties without a college campus.** (a) Reopening college campuses significantly increased the number of devices on college campuses ($\chi^2(28) = 1355.79$, $p < 0.001$). These increases persisted on college campuses (b) for at least eight weeks after reopening (In-person: $\chi^2(8) = 791.03$, $p < 0.001$; Online: $\chi^2(8) = 249.37$, $p < 0.001$) and were larger for campuses that reopened for primarily in-person teaching ($\chi^2(8) = 330.75$, $p < 0.001$). Reopening college campuses also increased the incidence of COVID-19 in the county, regardless of teaching modality using data from USAFacts (c; In-person $\chi^2(8) = 19.6$, p = 0.012; Online $\chi^2(8) = 18.8$, p = 0.016) and for in-person teaching using data from the CDC (d; In-person $\chi^2(8) = 18.5$, p = 0.018; Online $\chi^2(8) = 7.3$, p = 0.507]. After reopening there were significantly more cases resulting in hospitalization (e) associated with in-person, as opposed to online, reopening ($\chi^2(8) = 17.1$, $p = 0.029$). However, there were no significant differences in ICU utilization (f, $\chi^2(8) = 10.8$, $p = 0.215$) in the first eight weeks after reopening. During the first eight weeks, in-person teaching was not associated with a greater incidence of mortality, relative to online teaching ($\chi^2(4) = 13.8$, $p = 0.086$). Local transmission (h), measured by $R_t$, was significantly different from zero after reopening a college, regardless of teaching modality (In-person $\chi^2(8) = 40.4$, $p < 0.001$; Online $\chi^2(8) = 37.3$, $p < 0.001$). COVID-19 related data are from the CDC unless otherwise specified.

regardless of teaching modality, using national data from USAFacts (In-person $\chi^2(8) = 19.6$, $p = 0.012$; Online $\chi^2(8) = 18.8$, $p = 0.16$ Fig 1c), while in-person reopenings were accompanied by increasing disease incidence using CDC data (In-person $\chi^2(8) = 18.5$, $p = 0.018$; Online $\chi^2(8) = 7.28$, $p = 0.507$ Fig 1d). The increase in COVID-19 cases was not accompanied by increases in cases requiring hospitalization (In-person $\chi^2(8) = 5.3$, $p = 0.729$; Online $\chi^2(8) = 15.01$, $p = 0.059$, Fig 1e), ICU admissions (In-person $\chi^2(8) = 13.3$, $p = 0.100$; Online $\chi^2(8) = 8.30$, $p = 0.404$, Fig 1f), or that resulted in death (In-person $\chi^2(8) = 7.23$, $p = 0.512$; Online $\chi^2(8) = 10.7$, $p = 0.216$, Fig 1g). $R_t$ increased during the first eight weeks regardless of the teaching modality chosen by the college (In-person $\chi^2(8) = 30.4$, $p < 0.001$; Online $\chi^2(8) = 37.3$, $p < 0.001$, Fig 1h).

The weekly event study coefficients are presented in S3 Table.

## Difference-in-differences estimates

The reopenings lead to a cascade of indirect effects at the population level. To show this, we estimated a series of difference-in-difference models to estimate the effect of reopening a college campus on mobility and COVID-19 outcomes. We present the detailed results in the S4 Table, but describe the results below.

Reopening a college campus was associated with a 32.2 (95% CI: 29.3–35.1, $p < 0.001$) log point increase in the number of devices on campus, or approximately a 38.0% (95% CI: 34.0–42.0%) increase in movement on campus, from two weeks prior to the start of classes (S4 Table, column (1)). While we do find evidence of differential pre-trends ($\chi^2(5) = 186.18$, $p < 0.001$), we can bound the effect of these trends under various assumptions [35], in the post-period which yields statistically significant effects of college reopenings on mobility (see the Supplementary Information for details). The increase in movement was larger in schools that reopened for primarily in-person, as opposed to primarily online, instruction (38.8 [35.1–42.4] vs. 22.5 [19.1–25.9] log points; $\chi^2(1) = 166.04$, $p < 0.001$). The increase in mobility was also larger for colleges that had greater exposure to students from areas with high levels of COVID-19 incidence ($\chi^2(2) = 52.71$, $p < 0.001$) or had a greater share of college students, relative to population in the county ($\chi^2(2) = 243.4$, $p < 0.001$).

Using our difference-in-difference framework, we found that reopening a college was associated with a statistically significant increase of 4.9 cases per 100,000 people (95% CI: 2.9–6.9, $p < 0.001$; S4 Table, column (2)) using case data from USAFacts (or approximately 35% relative to the pre-period mean). The estimate was somewhat smaller using data from the CDC sample of counties (2.7 cases per 100,000 95% CI: 1.2–4.2, $p < 0.001$; S4 Table, column (3); 20% of the pre-period mean). In both cases, the results were robust to allowing for linear post-trends. The increase in COVID-19 incidence, using USAFacts data, was significantly different across terciles of exposure to COVID-19 due to student mobility ($\chi^2(2) = 8.9$, $p = 0.012$) with increased incidence in all three terciles. The increase in COVID-19 incidence was only significant in the third tercile across both USAFacts and CDC data and the difference across terciles of students share of the county population was significant in both specifications (USAFacts: $\chi^2(2) = 98.83$, $p < 0.001$; CDC: $\chi^2(2) = 27.48$, $p < 0.001$).

In our CDC data, we found no evidence of statistically significant increases in COVID-19 cases resulting in hospitalization ($p = 0.661$), ICU admission ($p = 0.057$), or death ($p = 0.391$) in the full sample, nor did we find evidence of differences by teaching modality. However, among the highest tercile of college student share, we observed statistically significant increases in cases resulting in hospitalization (0.148; 95% CI: 0.010–0.287; $p = 0.035$) and mortality (0.077; 95% CI: 0.001–0.154; $p = 0.047$).

The central epidemiological parameter governing a disease system's dynamics is the effective reproduction number ($R_t$). We estimated a significant increase in daily ($R_t$) around the time of reopening, consistent with an uptick in transmission. On average there was an increase in $R_t$ of 0.056 (95% CI: 0.025–0.087, $p < 0.001$). We note that $R_t$ did not significantly differ by the teaching method chosen for the campus ($\chi^2(1) = 0.23$, $p = 0.635$; S4 Table) but did decline over time ($\chi^2(2) = 42.95$, $p < 0.001$).

## Age-specific incidence

To observe the age-stratified dynamics, we explored age-specific incidence. Our analyses supported the conclusion that the shifts in age dynamics overtime likely resulted from college reopenings in Fig 2. The top panel (Fig 2a) demonstrates a clear shift, where we observe an increase in COVID-19 incidence in people ages 10–29, but not for any other age group ($\chi^2(5) = 67.2$, $p < 0.001$). However, the increase in these college-aged students was dramatic, with incidence among 10–19 year-olds rising by 8.0 (95% CI: 5.5–10.5, $p < 0.001$) cases per 100,000, or almost 80% of the pre-period mean, while for 20–29 year-olds the increase was 7.4 (95% CI: 4.6–10.1, $p < 0.001$) cases per 100,000 (38% of the pre-period mean). The second panel indicates that our estimates of the effect of reopening on hospitalizations by age group are too noisy to draw any inferences ($\chi^2(5) = 5.28$, $p = 0.382$). The third and fourth panels

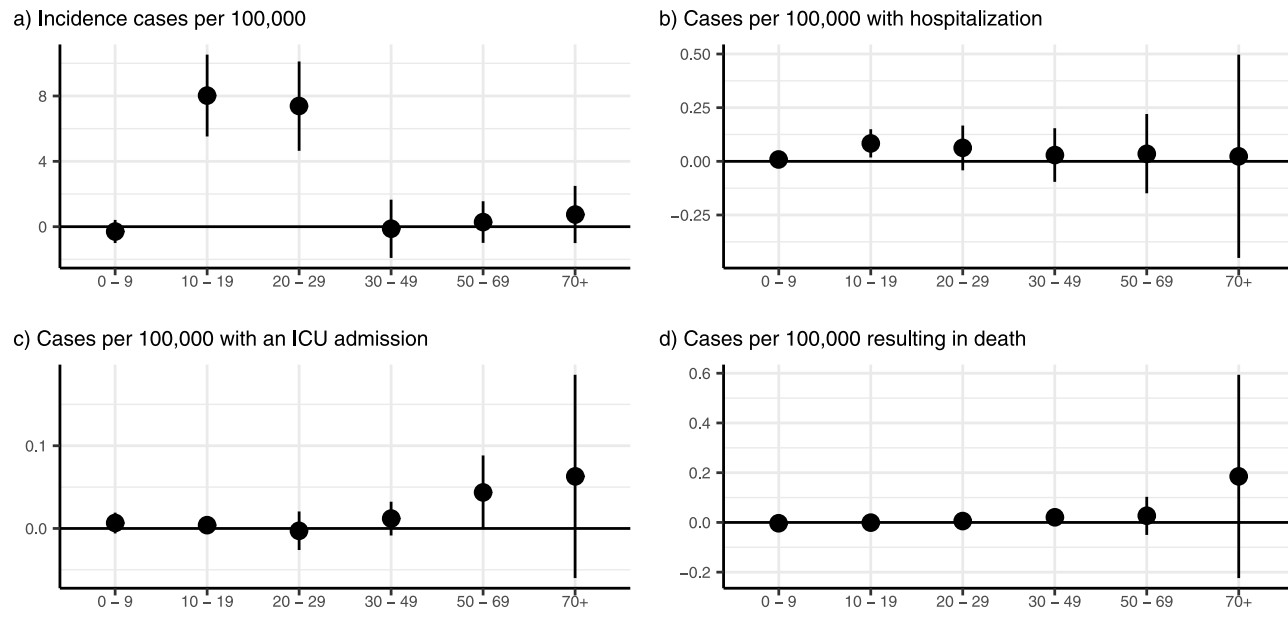

**Fig 2. Age-specific effects of college reopenings.** (a) demonstrates that the increase in the incidence of COVID-19 was isolated to people between 10 and 29 years of age, which encompasses most college-age individuals ($\chi^2(5) = 67.2$, $p < 0.001$). In the aggregate, hospitalization rates did not change differentially by age after reopening (b; $\chi^2(5) = 5.28$, $p = 0.382$). Similarly, we find no evidence of differential changes in the incidence of cases requiring ICU admission (c; $\chi^2(5) = 5.14$, $p = 0.398$). We did not find any age-specific increases in mortality due to COVID-19, although these results were imprecisely estimated (d; $\chi^2(5) = 7.57$, $p = 0.181$). Figure plots point estimates and 95% confidence intervals. Point estimates and standard errors are available in S5 Table.

demonstrate that there were no statistically significant age-specific increase in cases requiring an ICU admission or resulting in death following a campus reopening.

While the data appear to paint a clear picture, it is possible that several mechanisms may yield a similar age-specific profile of cases. Thus, in the SI, we test these observations. We show age-specific event studies (S3 Fig) for our four age-specific outcomes. These event studies clearly demonstrate that all age groups were trending similarly prior to the reopening, except for hospitalizations (cases: $\chi^2(20) = 13.76$, $p = 0.843$; hospitalizations: $\chi^2(20) = 37.62$, $p = 0.010$; ICU $\chi^2(20) = 26.61$, $p = 0.147$; and deaths ($\chi^2(20) = 23.35$, $p = 0.272$). COVID-19 incidence rose beginning in the week campuses reopened and remained elevated subsequently, with statistically significant increases for at least four weeks after reopening ($\chi^2(20) = 92.58$, $p < 0.001$). We also evidence of changes in the post period by age group for hospitalizations ($\chi^2(20) = 34.21$, $p = 0.024$) and deaths ($\chi^2(20) = 38.97$, $p = 0.007$) after reopening.

## Differential effects by teaching modality

We disaggregated teaching modality into more granular categories to explore potential differences across teaching methods. Our results demonstrate that campuses that reopened with a greater emphasis on in-person teaching were associated with larger increases in mobility ($\chi^2(4) = 331.80$, $p < 0.001$)) and incident COVID-19 cases (USAFacts: $\chi^2(4) = 14.20$, $p = 0.007$; CDC: $\chi^2(4) = 2.52$, $p = 0.641$) (Fig 3). However, these results are tempered since we rejected null hypothesis of no differential pre-trends across counties in the USAFacts data ($\chi^2(16) = 38.04$, $p = 0.001$). Event studies for these outcomes by teaching modality are available in the S4 Fig.

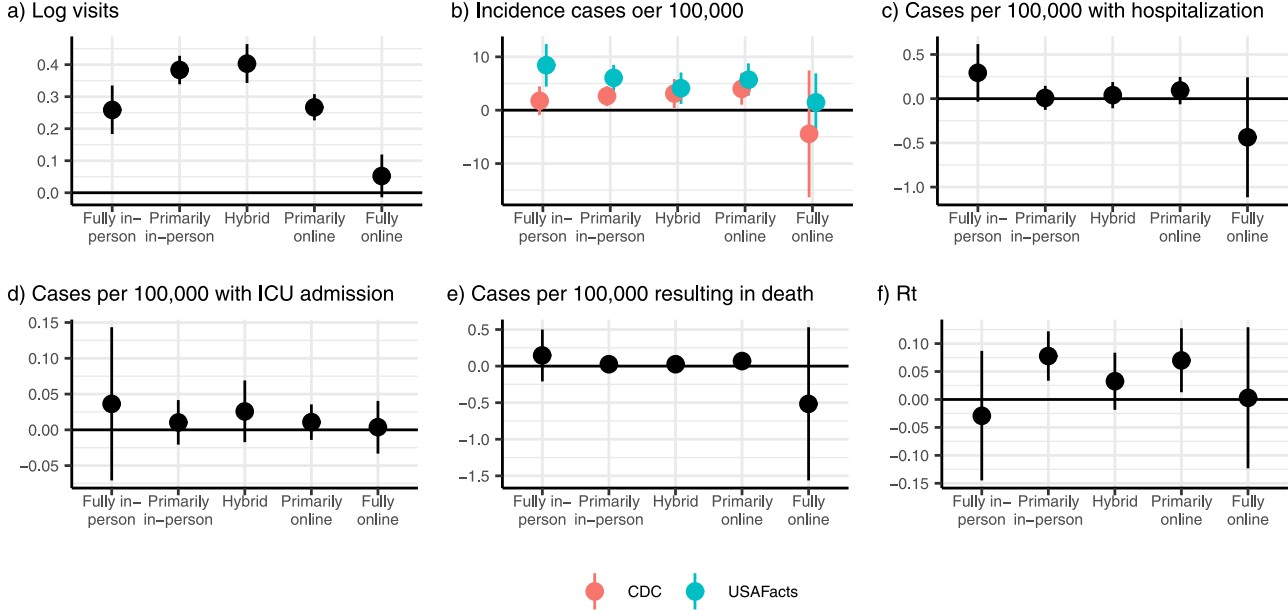

**Fig 3. Differential effects of reopening college campuses by expanded teaching modality.** (a) Campuses that reopened for "Primarily in-person" and "Hybrid" teaching had the largest increase in devices on campus following reopening, while the increase in visitors was significantly smaller for fully online reopenings ($\chi^2(4) = 331.80$, $p < 0.001$). All reopenings except "Fully Online" were associated with a significant increase in COVID-19 cases after reopening (b) using incidence data from USAFacts ($\chi^2(4) = 14.20$, $p = 0.007$). There were no statistically significant adverse effects of reopening a college campus by teaching modality for hospitalizations (c), ICU admissions (e), or mortality following a reopening (e). For two groups–"primarily in-person" and "primarily online"–we found evidence of increases in $R_t$ (f; $\chi^2(4) = 12.06$, $p = 0.017$).

## Discussion

Schools and universities welcomed millions of students back to campus for Fall 2021 while vaccine uptake stagnated [37] and many people on campuses and in surrounding communities remain vaccine hesitant. Therefore, revisiting the effects of mass movement events into campuses and how they affect local and surrounding communities is crucial. Our results provide a quantitative evaluation of mobility patterns during periods of reactive college closure and reopening strategies and highlight their impact in shaping social interactions of the college and surrounding communities. We found that college policies induced marked changes in the overall number of daily mobility interactions. Our findings demonstrate that re-opening a college was associated with an increase in the number of cellular devices on campus (a dramatic increase in population size) after classes resumed for all teaching modalities, although the increase in mobility is larger for in-person as opposed to online teaching. We unequivocally showed that re-opening a college significantly increased the incidence of COVID-19 in the county. In counties that reopened for in-person teaching teaching we also demonstrated that reopening increased the incidence of cases resulting in an ICU admission. We also demonstrated that counties containing colleges that drew students from areas with higher COVID-19 incidence experienced significantly larger increases in COVID-19 incidence following campus reopening. This is likely induced by the dramatic increase in the number of contacts of students with each other on the campuses and with the surrounding communities.

To contextualize our findings, there are 238.0 million Americans in the 786 counties that contain a college campus in our sample. Our results demonstrate that reopening college campuses resulted in an additional 11,500 (11626 [95% CI: 6945–16310]) cases of COVID-19 per day. This estimate is larger than the aggregate number of cases reported on the New York

Times case tracker which reported more than 397,000 cases as of December 11 2020 [38], but the New York Times tracker excludes spillover effects into the community.

However, because of the nature of the cases reports data, we were unable to disentangle how many of the cases we measure as our outcome are "imported" (student arrivals) and how many are local transmissions from the students. Further, asymptomatic cases were only identified if testing was done on campus regardless of symptoms. Nevertheless, our results are inconsistent with large numbers of "imported" cases since an imported case would lead to an increase in COVID-19 cases contemporaneously with any increase in mobility, while we observed a one-week lag between peak mobility and the peak change in COVID-19 incidence, when cases are assigned based on symptom onset.

We did not quantify potential spillovers to the communities surrounding campuses, as these effects would require college-level incidence data, which are not consistently collected. However, using age-specific data, we were able to demonstrate that most of the increase in COVID-19 incidence arose among college-aged students (ages 10–29).

Additional work is necessary to identify the optimal reopening- closure policies (e.g., lengths) and under which circumstances specific policies are cost-effective. However, evaluating the effectiveness of specific mitigation measures taken by colleges, especially the ways in which colleges have reacted to the initial increases in cases with strong countermeasures, was beyond the scope of this initial study and remain priorities for future studies. Similarly, we were unable to test what has occurred once colleges change decisions, such as changing instructional modes temporarily or encouraging students to return home [9] since these changes were reactions to rapidly increasing case counts [39].

While we only directly demonstrate that college campuses that were more heavily exposed to COVID-19 lead to larger increases in incidence, our results also indicate that sending students home from colleges due to high COVID-19 incidence is likely to lead to increased COVID-19 incidence in students' home communities since the same exposure mechanism would run in reverse. Public health officials have also raised these concerns, some of whom have publicly opposed closing dormitories, even after a college or university transitioned to online education [40]. Further research on the effects of sending students home is needed to understand the risks and benefits of closing residence halls.

The nature of our data limits our results. Our mobility analysis relies on observing cellular GPS signals and these devices may not always report their location. In addition, it is unlikely that devices correspond in a one-to-one manner with people since college students may have more than one device (a phone and a cell-enabled tablet) that provide data under distinct identifiers. Second, we are unable to measure cases among college students vs. others in the county community, beyond using the age of the individual. Third, our mobility measure does not take account of students who may live in off-campus housing and take classes online. Fourth, there is some evidence of differential pre-trends, which may affect our point estimates. Using methods that are robust to violations of the parallel trends assumption [35] we find that our point estimates are consistently included in the identified sets for various plausible violations of the parallel trends assumptions. Fifth, our classification of counties into various treatment groups is based on the first and largest campus, but in the Supplementary Information we demonstrate that our results continue to hold when we use counties with no or one campus or university.

## Conclusion

Our results demonstrate the essential role that mixing and mobility play in seeding COVID-19 in the community and the role that congregate living settings play in providing a fertile ground

for COVID-19 to expand. For example, these results highlight the role that nursing homes and prisons play in the COVID-19 pandemic and complement existing research on cross-nursing home linkages and COVID-19 incidence [1]. While we expect that continued testing on college campuses and current vaccination efforts will mitigate some of the effects we observed, the rate of vaccination remains low, particularly among college-age individuals [41], and the majority of colleges did not engage in high-quality testing regimes in the 2020–2021 academic year [42]. Furthermore, the emergence of new variants, such as omicron, that exhibit an ability to evade immune responses from previous infection with COVID-19 or vaccination [43–45] highlights the importance of a "defense-in-depth" strategy for colleges and universities that includes adjusting teaching modalities to limit the spread of COVID-19. Our results demonstrate the effectiveness of strategies such as on-line teaching against more severe health consequences, versus continuing with business as usual.

Our analysis is a good step towards building a framework to map mobility to contacts (as COVID-19 era contact matrices become available [46]), in the analysis of airborne infectious diseases. As such, our framework has a much wider scope than the study of COVID-19 related college policies in one specific region. Our findings are critical in the context of adapting public health management strategies, as they consider additional strategies to mitigate disease burden and decrease transmission. The effects of college reopenings are also informative for outbreak management in other communal settings, including nursing homes and prisons, both of which have been particularly hard hit by the COVID-19 pandemic.

Despite all data limitations, our study, provides (i) empirical evidence about changes in "behavior" (mobility surges) of the population during the implementation of the school-reopening strategies, (ii) a multi pronged approach to estimate mobility patterns and evaluate effects on the spread of infectious diseases with an unique degree of detail and (iii) tools for evidence-based decision-making beyond evaluating college reopening strategies.

## Supporting information

**S1 Fig. Geographic distribution of counties that are included in the CDC sample.** Counties were included in the CDC sample if the ratio of cumulative cases during our study period in the CDC line files relative to USAFacts was greater than 0.95 and the correlation in the 30-day cumulative case count time series was greater than 0.95.
(PDF)

**S2 Fig. Geographic distribution of colleges and universities in sample by teaching modality.** Colleges and universities are more prevalent in the eastern half of the United States, while colleges in the western half were more likely to reopen for online teaching.
(PDF)

**S3 Fig. Age-specific event studies.** COVID-19 incidence rose for the 10-19 and 20-29 year old age groups following the resumption of classes, although the increase faded over time for both age groups (a), while cases requiring hospitalization (b) and ICU care (c) followed less precisely estimated patterns with a decreasing time trend throughout the study period. Mortality due to COVID-19 was stable throughout the study period as well for all age groups (d).
(PDF)

**S4 Fig. Teaching modality-specific event studies.** On-campus mobility increased for all teaching modalities (a), while COVID-19 incidence rose for "Fully in-person" ($\chi^2(8) = 30.16$, $p < 0.001$), "Primarily in-person" ($\chi^2(8) = 27.79$, $p < 0.001$), and "Primarily online" ($\chi^2(8) = 21.25$, $p = 0.007$) teaching modalities (b). CDC data (c) indicated that there were increases in COVID-19 incidence for "Hybrid" reopenings ($\chi^2(8) = 26.52$, $p < 0.001$). Hospitalization rates

rose following "Fully in-person" ($\chi^2(8) = 16.03$, $p = 0.042$) reopenings (d). ICU admissions increased following reopening for "Hybrid" ($\chi^2(8) = 21.43$, $p = 0.006$) reopenings (e). Deaths did not increase for any reopening modality (f). $R_t$ was significantly different following reopening for "Primarily in-person" ($\chi^2(8) = 23.52$, $p = 0.003$), "Hybrid" ($\chi^2(8) = 25.66$, $p = 0.001$), and "Primarily online" ($\chi^2(8) = 25.82$, $p = 0.001$) reopenings (g).
(PDF)

**S5 Fig. Geographic distribution of teaching assignments.** Counties were assigned on the basis of the earliest and, if necessary, largest college or university in each county.
(PDF)

**S1 Appendix. Additional methods.**
(PDF)

**S2 Appendix. Robustness checks and alternative specifications.**
(PDF)

**S1 Table. Teaching modalities across counties.**
(PDF)

**S2 Table. Summary statistics on the sample.**
(PDF)

**S3 Table. Event study coefficients.**
(PDF)

**S4 Table. Difference-in-difference regressions demonstrating the effect of reopening college campuses on mobility and COVID-19 incidence and sequelae.**
(PDF)

**S5 Table. Age-specific difference-in-differences demonstrates that the increase in incidence of COVID-19 was isolated to people between 10 and 29 years of age, which encompasses college-age individuals.**
(PDF)

**S6 Table. Age-specific means of the dependent variables.**
(PDF)

**S7 Table. Robustness checks discussed in section S2 Appendix.**
(PDF)

**S8 Table. Weekly and daily pre-trend tests.**
(PDF)

## Acknowledgments

We would like to thank the University of North Carolina at Chapel Hill and the Research Computing group for providing computational resources and support that have contributed to these research results. Brant Callaway provided helpful advice on implementing the DiD estimator.

## Author Contributions

**Conceptualization:** Martin S. Andersen, Ana I. Bento, Anirban Basu, Christopher R. Marsicano, Kosali I. Simon.

**Data curation:** Martin S. Andersen.

**Formal analysis:** Martin S. Andersen.

**Project administration:** Martin S. Andersen.

**Resources:** Martin S. Andersen.

**Visualization:** Ana I. Bento.

**Writing – original draft:** Martin S. Andersen, Ana I. Bento.

**Writing – review & editing:** Anirban Basu, Christopher R. Marsicano, Kosali I. Simon.

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
