## [Decision Letter · Decision Letter 0]

5 May 2022

PONE-D-22-02504College openings in the United States increase mobility and COVID-19 incidencePLOS ONE

Dear Dr. Andersen,

Thank you for submitting your manuscript to PLOS ONE. After careful consideration, we feel that it has merit but does not fully meet PLOS ONE’s publication criteria as it currently stands. Therefore, we invite you to submit a revised version of the manuscript that addresses the points raised during the review process.

We look forward to receiving your revised manuscript.

Kind regards,

Gabriel A. Picone

Academic Editor

PLOS ONE

Journal Requirements:

When submitting your revision, we need you to address these additional requirements. 1. Please ensure that your manuscript meets PLOS ONE's style requirements, including those for file naming. The PLOS ONE style templates can be found at https://journals.plos.org/plosone/s/file?id=wjVg/PLOSOne_formatting_sample_main_body.pdf and https://journals.plos.org/plosone/s/file?id=ba62/PLOSOne_formatting_sample_title_authors_affiliations.pdf 2. Please provide additional details regarding participant consent. In the ethics statement in the Methods and online submission information, please ensure that you have specified (1) whether consent was informed and (2) what type you obtained (for instance, written or verbal, and if verbal, how it was documented and witnessed). If your study included minors, state whether you obtained consent from parents or guardians. If the need for consent was waived by the ethics committee, please include this information. If you are reporting a retrospective study of medical records or archived samples, please ensure that you have discussed whether all data were fully anonymized before you accessed them and/or whether the IRB or ethics committee waived the requirement for informed consent. If patients provided informed written consent to have data from their medical records used in research, please include this information Once you have amended this/these statement(s) in the Methods section of the manuscript, please add the same text to the “Ethics Statement” field of the submission form (via “Edit Submission”). For additional information about PLOS ONE ethical requirements for human subjects research, please refer to http://journals.plos.org/plosone/s/submission-guidelines#loc-human-subjects-research. 3. In your Data Availability statement, you have not specified where the minimal data set underlying the results described in your manuscript can be found. PLOS defines a study's minimal data set as the underlying data used to reach the conclusions drawn in the manuscript and any additional data required to replicate the reported study findings in their entirety. All PLOS journals require that the minimal data set be made fully available. For more information about our data policy, please see http://journals.plos.org/plosone/s/data-availability. "Upon re-submitting your revised manuscript, please upload your study’s minimal underlying data set as either Supporting Information files or to a stable, public repository and include the relevant URLs, DOIs, or accession numbers within your revised cover letter. For a list of acceptable repositories, please see http://journals.plos.org/plosone/s/data-availability#loc-recommended-repositories. Any potentially identifying patient information must be fully anonymized. Important: If there are ethical or legal restrictions to sharing your data publicly, please explain these restrictions in detail. Please see our guidelines for more information on what we consider unacceptable restrictions to publicly sharing data: http://journals.plos.org/plosone/s/data-availability#loc-unacceptable-data-access-restrictions. Note that it is not acceptable for the authors to be the sole named individuals responsible for ensuring data access. We will update your Data Availability statement to reflect the information you provide in your cover letter. 4. We note that Supplementary Figures 1 to 3 in your submission contain [map/satellite] images which may be copyrighted. All PLOS content is published under the Creative Commons Attribution License (CC BY 4.0), which means that the manuscript, images, and Supporting Information files will be freely available online, and any third party is permitted to access, download, copy, distribute, and use these materials in any way, even commercially, with proper attribution. For these reasons, we cannot publish previously copyrighted maps or satellite images created using proprietary data, such as Google software (Google Maps, Street View, and Earth). For more information, see our copyright guidelines: http://journals.plos.org/plosone/s/licenses-and-copyright. We require you to either (1) present written permission from the copyright holder to publish these figures specifically under the CC BY 4.0 license, or (2) remove the figures from your submission: a. You may seek permission from the original copyright holder of Supplementary Figures 1 to 3 to publish the content specifically under the CC BY 4.0 license.   We recommend that you contact the original copyright holder with the Content Permission Form (http://journals.plos.org/plosone/s/file?id=7c09/content-permission-form.pdf) and the following text:“I request permission for the open-access journal PLOS ONE to publish XXX under the Creative Commons Attribution License (CCAL) CC BY 4.0 (http://creativecommons.org/licenses/by/4.0/). Please be aware that this license allows unrestricted use and distribution, even commercially, by third parties. Please reply and provide explicit written permission to publish XXX under a CC BY license and complete the attached form.” Please upload the completed Content Permission Form or other proof of granted permissions as an ""Other"" file with your submission. In the figure caption of the copyrighted figure, please include the following text: “Reprinted from [ref] under a CC BY license, with permission from [name of publisher], original copyright [original copyright year].” b. If you are unable to obtain permission from the original copyright holder to publish these figures under the CC BY 4.0 license or if the copyright holder’s requirements are incompatible with the CC BY 4.0 license, please either i) remove the figure or ii) supply a replacement figure that complies with the CC BY 4.0 license. Please check copyright information on all replacement figures and update the figure caption with source information. If applicable, please specify in the figure caption text when a figure is similar but not identical to the original image and is therefore for illustrative purposes only.The following resources for replacing copyrighted map figures may be helpful: USGS National Map Viewer (public domain): http://viewer.nationalmap.gov/viewer/The Gateway to Astronaut Photography of Earth (public domain): http://eol.jsc.nasa.gov/sseop/clickmap/Maps at the CIA (public domain): https://www.cia.gov/library/publications/the-world-factbook/index.html and https://www.cia.gov/library/publications/cia-maps-publications/index.htmlNASA Earth Observatory (public domain): http://earthobservatory.nasa.gov/Landsat: http://landsat.visibleearth.nasa.gov/USGS EROS (Earth Resources Observatory and Science (EROS) Center) (public domain): http://eros.usgs.gov/#Natural Earth (public domain): http://www.naturalearthdata.com/

Reviewers' comments:

Reviewer's Responses to Questions

**Comments to the Author**

1. Is the manuscript technically sound, and do the data support the conclusions?

Reviewer #1: Yes

Reviewer #2: Yes

2. Has the statistical analysis been performed appropriately and rigorously? 

Reviewer #1: Yes

Reviewer #2: Yes

3. Have the authors made all data underlying the findings in their manuscript fully available?

Reviewer #1: Yes

Reviewer #2: Yes

4. Is the manuscript presented in an intelligible fashion and written in standard English?

Reviewer #1: Yes

Reviewer #2: Yes

5. Review Comments to the Author

Reviewer #1: The authors have an innovative paper on college reopening in Fall 2020. They show significant effects on mobility -- almost by definition mechanical effects since college campus are nearly vacant in summer -- using SafeGraph data. They also show significant increases in cases -- somewhat invariant to teaching modality -- at the county level relative to non-college counties. The paper does important diagnostics with generalized difference-in-differences / event studies. The paper is well written.

There are several important avenues to explore in a revision.

First, and most importantly, the most noteworthy and credible results are that college reopenings increased COVID-19 cases in the community (county). Yet the parameterization of "reopening" is challenging for two key reasons. One reason is that colleges vary tremendously in size (some large public universities have 50,000+ students while small liberal arts colleges might have 1,000 or so students), so the dosage within a community varies tremendously. Related, colleges size within a community matters a lot too -- colleges are a dominant shift in population in many "college towns" but may be insignificant in large urban centers. The second reason is that the authors focus on the first college reopening within the community. This is likely uncontroversial for small rural communities where there is one college, but is certainly problematic for analyzing COVID-19 spread within large urban centers (which can have dozens of colleges). There are some clear suggestions that can address these specific concerns. We'd expect the models of COVID cases, hospitalizations, etc., to have stronger effects in communities where student populations are an important part of county populations (e.g., I'm guess more rural settings). In addition, it appears that in "college counties" there are on average 2 colleges per county. But that almost certainly masks many rural counties with just 1 college, and some highly urban counties with 30+ colleges. By focusing on counties with a small number of colleges, the "first opening" measure would be more convincing.

Second, there was little mention of K-12 reopenings for Fall 2020. This must be addressed empirically as well; some states (e.g., Florida, Texas, etc.) were very aggressive about reopening public schools, while other states stayed online in Fall 2020. In principle, college reopenings in states (or counties) where K-12 remained online would provide the most convincing evidence on colleges; otherwise, there are natural concerns about similar timing of these two potential spreaders of COVID. It might be the case that the "no college" counties serve as a control for public school reopenings, but the case would need to be convincingly made.

Third, in the introduction the authors say "As we approach Fall 2021". They should update. The conclusion seems to be updated to 2022.

Overall, this is a terrific paper on a controversial topic that generates extreme reactions. The authors' findings make a lot of sense (although some of the extreme outcomes like mortality seem to increase relatively quickly compared to the lag we might expect from cases -- especially if spread starts among the young). However, the exploration I suggest isn't just "robustness" or footnote suggestions (in my view). Trying to characterize college reopenings in large urban areas is really hard given the multiplicity of colleges and differences in start dates. And even if colleges are areas of extreme transmission, some colleges are really small.

Reviewer #2: This study contributes to an important debate on the role of college reopening policies in mitigating the transmission of COVID-19 infections. The authors use data from a variety of sources to evaluate the association between college re-openings and mobility and COVID-19 related outcomes. The study is well done. I have some minor suggestions:

• In the introduction (paragraph 3), the authors write “As we approach Fall 2021, with expected mass movement events in the US …” Given that we are now in Spring 2022, this discussion should be updated to reflect the current situation. Many of these arguments remain relevant today.

• Is the natural log of county population the only variable that is included in the difference-in-differences model? Clarify whether other covariates are included in the propensity score.

• The discussion and/or conclusion sections should acknowledge the presence of pre-trends in mobility, which limits the causal interpretation of the estimates. Clarify that a causal interpretation is only valid under the assumption that any pre-trends are fully captured by linear trends.

• Given that the sample includes only 70% of four-year colleges, are counties with colleges not included in the C2i data dropped from the sample? In other words, does the control group include counties with colleges for which you do not have opening dates? This should be clarified.

6. PLOS authors have the option to publish the peer review history of their article (what does this mean?). If published, this will include your full peer review and any attached files.

Reviewer #1: No

Reviewer #2: No

---

## [Decision Letter · Decision Letter 1]

27 Jul 2022

College openings in the United States increase mobility and COVID-19 incidence

PONE-D-22-02504R1

Dear Dr. Andersen,

We’re pleased to inform you that your manuscript has been judged scientifically suitable for publication and will be formally accepted for publication once it meets all outstanding technical requirements.

Kind regards,

Gabriel A. Picone

Academic Editor

PLOS ONE

Additional Editor Comments (optional):

Reviewers' comments:

Reviewer's Responses to Questions

**Comments to the Author**

1. If the authors have adequately addressed your comments raised in a previous round of review and you feel that this manuscript is now acceptable for publication, you may indicate that here to bypass the “Comments to the Author” section, enter your conflict of interest statement in the “Confidential to Editor” section, and submit your "Accept" recommendation.

Reviewer #1: All comments have been addressed

Reviewer #2: All comments have been addressed

2. Is the manuscript technically sound, and do the data support the conclusions?

Reviewer #1: Yes

Reviewer #2: Yes

3. Has the statistical analysis been performed appropriately and rigorously? 

Reviewer #1: Yes

Reviewer #2: Yes

4. Have the authors made all data underlying the findings in their manuscript fully available?

Reviewer #1: Yes

Reviewer #2: Yes

5. Is the manuscript presented in an intelligible fashion and written in standard English?

Reviewer #1: Yes

Reviewer #2: Yes

6. Review Comments to the Author

Reviewer #1: The authors have addressed all my concerns. I appreciate the care taken in this study and will use it in my classes.

Reviewer #2: (No Response)

7. PLOS authors have the option to publish the peer review history of their article (what does this mean?). If published, this will include your full peer review and any attached files.

Reviewer #1: No

Reviewer #2: No

---

## [Editor Report · Acceptance letter]

19 Aug 2022

PONE-D-22-02504R1 

College openings in the United States increase mobility and COVID-19 incidence 

Dear Dr. Andersen:

I'm pleased to inform you that your manuscript has been deemed suitable for publication in PLOS ONE. Congratulations! Your manuscript is now with our production department. 

Kind regards, 

on behalf of

Dr. Gabriel A. Picone 

Academic Editor

PLOS ONE